# Comparative Effects of Acetate- and Citrate-Based Dialysates on Dialysis Dose and Protein-Bound Uremic Toxins in Hemodiafiltration Patients: Exploring the Impact of Calcium and Magnesium Concentrations

**DOI:** 10.3390/toxins16100426

**Published:** 2024-10-01

**Authors:** Diana Rodríguez-Espinosa, Elena Cuadrado-Payán, Naira Rico, Mercè Torra, Rosa María Fernández, Miquel Gómez, Laura Morantes, Gregori Casals, Maria Rodriguez-Garcia, Francisco Maduell, José Jesús Broseta

**Affiliations:** 1Nephrology and Renal Transplantation, Hospital Clínic de Barcelona, 08036 Barcelona, Spain; dmrodriguez@clinic.cat (D.R.-E.); ecuadrado@clinic.cat (E.C.-P.); mgomezu@recerca.clinic.cat (M.G.); morantes@clinic.cat (L.M.); fmaduell@clinic.cat (F.M.); 2Biochemistry and Molecular Genetics Department-CDB, Hospital Clínic de Barcelona, 08036 Barcelona, Spain; nrico@clinic.cat (N.R.); mtorra@clinic.cat (M.T.); rmfernandez@clinic.cat (R.M.F.); casals@clinic.cat (G.C.); mrodriguezg@clinic.cat (M.R.-G.)

**Keywords:** hemodialysis, citrate, acetate, uremic toxins, dialysis efficiency, protein-bound toxins, p-cresyl sulfate, indoxyl sulfate

## Abstract

Modern hemodialysis employs weak acids as buffers to prevent bicarbonate precipitation with calcium or magnesium. Acetate, the most used acid, is linked to chronic inflammation and poor dialysis tolerance. Citrate has emerged as a potential alternative, though its effect on dialysis efficiency is not clear. This study aims to compare the efficacy of acetate- and citrate-based dialysates, focusing on protein-bound uremic toxins and dialysis doses. This single-center prospective crossover study includes prevalent patients participating in a thrice-weekly online hemodiafiltration program. Four dialysates were tested: two acetate-based (1.25 and 1.5 mmol/L calcium) and two citrate-based (1.5 mmol/L calcium with 0.5 and 0.75 mmol/L magnesium). Pre- and post-dialysis blood samples of eighteen patients were analyzed for urea, creatinine, p-cresyl sulfate, indoxyl sulfate, and albumin. Statistical significance was assessed using paired *t*-tests and repeated measures of ANOVA. There were no significant differences in dialysis dose (Kt), urea, creatinine, or indoxyl sulfate reduction ratios between acetate- and citrate-based dialysates. However, a significant decrease in the reduction ratio of p-cresyl sulfate was observed with the acetate dialysate containing 1.25 mmol/L calcium and the citrate dialysate with 0.5 mmol/L magnesium compared to the acetate dialysate containing 1.5 mmol/L calcium and the citrate dialysate with 0.75 mmol/L magnesium (51.56 ± 4.75 and 53.02 ± 4.52 vs. 65.25 ± 3.38 and 58.66 ± 4.16, *p* 0.007). No differences in dialysis dose were found between acetate- and citrate-based dialysates. However, citrate dialysates with lower calcium and magnesium concentrations may reduce the albumin displacement of p-cresyl sulfate. Further studies are needed to understand the observed differences and optimize the dialysate composition for the better clearance of protein-bound uremic toxins.

## 1. Introduction

Modern hemodialysis requires using a weak acid to act as a buffer and avoid precipitation in the salt form of bicarbonate with calcium or magnesium in the dialysate [1]. Nowadays, the most frequently and commonly used buffer is acetate; however, this compound is associated with chronic inflammation [2,3], inadequate tolerance [4,5], and oxidative stress [6]. In this scenario, alternatives have emerged to replace acetate; among these, the most used one is citrate. The main benefit is that citrate offers acetate-free dialysis. However, there are beneficial effects reported, such as better tolerance to dialysis [4,7,8,9], less coagulation of the dialyzer [10], and a potential reduction in vascular calcification [11,12]. The citrate in the dialysis fluid binds plasma calcium and magnesium at a ratio of 1:1.5 to form calcium or magnesium citrate, Ca_3_(C_6_H_5_O_7_)_2_ and Mg_3_(C_6_H_5_O_7_)_2_, respectively [1], reducing the available amount of those divalent cations and preventing full activation of the coagulation cascade with the consequent increase in the maintenance of the dialyzer’s useful surface area, which should translate into better clearance and greater dialysis efficiency. However, the evidence of the latter is not clear, especially regarding single-use dialyzers and hemodiafiltration. Positive studies have been performed with high-flux hemodialysis [13,14,15] or dialyzer reuse [16,17]. More recent studies performed on a mixed population of hemodiafiltration (HDF) or high-flux hemodialysis (HF-HD) and single-use dialyzers have found no differences [9,11,18].

The current method to determine dialysis efficiency is a model proposed by Gotch and Sargent [19] based on the clearance of urea (Kt/V). Urea is a small molecule and an end product of protein catabolism [20], which is easy to measure and corresponds to the body’s total water volume [21]. There are limitations to this method, particularly concerning the V section of the equation, which includes factors like body weight, mass index, sex, and body surface area, all of which are independently associated with survival outcomes in dialysis patients [22,23,24,25]. Hence, many authors suggest that Kt is a more accurate determinant of the dialysis dose [26,27,28]. Beyond these concerns, the use of urea itself is controversial, as it may not fully represent the depurative capability of dialysis. Several studies have shown lower mortality rates with convective dialysis techniques, which can achieve similar urea clearance but are more effective at removing larger molecules [29,30,31]. Additionally, molecules with a high percentage of protein binding may not be adequately characterized by urea clearance alone and can become valuable parameters in monitoring dialysis efficiency.

Protein-bound uremic toxins, such as p-cresyl and indoxyl sulfate, are challenging to dialyze, given their affinity for albumin [32,33]. Even though the addition of medications is being studied as a displacer of uremic toxins from albumin [34,35,36,37], there are naturally occurring substances that compete for their binding site, such as calcium and magnesium [38,39,40], which are usually chelated with the use of citrate and may, therefore, interfere with the clearance of these toxins. 

Given that our group detected differences in the reduction ratios of p-cresyl sulfate in a previous investigation comparing acetate to citrate dialysates [41], the aim of this study is to determine whether these dialysates with different calcium and magnesium concentrations are used in the reduction ratios of protein-bound uremic toxins and their effect on the classic dialysis dose parameter determined by Kt.

## 2. Results

Eighteen patients with a median age of 80 (69–83) participated; 13 (72%) were male. Twelve had an arteriovenous fistula (AVF) as vascular access (66.7%), and the remaining used a tunneled central venous catheter. All dialysis sessions proceeded smoothly without incident, maintaining the same heparin dose and avoiding significant coagulation in the extracorporeal circuit. 

### 2.1. Citrate vs. Acetate

The results obtained with citrate-based dialysates were compared to those with acetate-based dialysates. Analysis revealed no statistically significant differences in dialysis dose, expressed as Kt, between the two types of dialysates. Additionally, no significant differences were observed in the reduction ratios (RR) of blood urea nitrogen (BUN), creatinine, p-cresyl sulfate, and indoxyl sulfate between acetate and citrate (see Table 1).

### 2.2. Analysis between Different Cation Concentrations 

Given the differences in calcium and magnesium between dialysates, we also analyzed both acetate- and citrate-buffered dialysates against each other. There were no statistically significant differences between SmartBag^®^ 211.25 (calcium of 1.25 mmol/L) and 211.5 (calcium of 1.5 mmol/L) in KT (70.1 ± 6.1 vs. 70.97 ± 7.2, *p* = 0.32), BUN RR (84.6 ± 3.1 vs. 85.23 ± 4, *p* = 0.33), nor creatinine RR (78.1 ± 4.4 vs. 77.8 ± 4.7, *p* = 0.71). There were also no differences between SmartBag^®^ CA 211.5 (magnesium of 0.5 mmol/L) and CA 211.5-0.75 (magnesium 0.75 mmol/L) in the KT (71.6 ± 7.5 vs. 71.5 ± 6.5, *p* = 0.89), BUN RR (85.6 ± 3 vs. 84.8 ± 2.9, *p* = 0.17), nor creatinine RR (78.9 ± 4.2 vs. 77.9 ± 4.7, *p* = 0.18).

### 2.3. Four Dialysates against Each Other

There were no statistically significant differences in the p-cresyl sulfate median pre-dialysis values between SmartBag^®^ 211.25 (43,308 ng/mL, 32,095–80,991 ng/mL), 211.5 (56,728 ng/mL, 42,578–89,815 ng/mL), CA 211.5 (50,074 ng/mL, 39,417–76,838 ng/mL), and CA 211.5-0.75 (54,813 ng/mL, 35,332–80,281 ng/mL) (*p* = 0.372), nor in the indoxyl-sulfate median pre-dialysis values between SmartBag^®^ 211.25 (51,024 ng/mL, 38,264–63,785 ng/mL), 211.5 (53,004 ng/mL, 41,769–64,240 ng/mL), CA 211.5 (55,615 ng/mL, 42,583–68,648 ng/mL) and CA 211.5-0.75 (53,587 ng/mL, 42,435–64,740 ng/mL) (*p* = 0.6).

Results from all four dialysates were analyzed. No differences were found in Kt, urea, creatinine, or indoxyl-sulfate (Figure 1).

A significant difference was noted in the reduction ratio of p-cresyl sulfate between the four studied dialysates (see Table 2 for details). Specifically, the reduction ratio was higher with the use of SmartBags 211.5 and CA 211.5-0.75, which both had a calcium concentration of 1.5 mmol/L; the former was acetate-based, while the latter was citrate-based with a higher magnesium concentration of 0.75 mmol/L. 

Given the significant difference in the p-cresyl results, we further analyzed this variable by examining the adjusted marginal means differences and found significant examples between SmartBag^®^ 211.5 with SmartBag^®^ 211.25 and SmartBag^®^ CA 211.5. There were no significant differences when compared to SmartBag^®^ CA 211.5-0.75 (see Table 3 for further information).

## 3. Discussion

There was a significantly greater reduction ratio of p-cresyl sulfate when patients used the acetate dialysate with a standard calcium concentration of 1.5 mmol/L than with 1.25 mmol/L or with the citrate dialysate with a calcium concentration of 1.5 mmol/L and a magnesium concentration of 0.5 mmol/L. There were no significant differences when compared to the magnesium-supplemented citrate dialysate (0.75 mmol/L).

Several studies have analyzed the depurative capacity of citrate vs. acetate, yielding mixed results. Two studies comparing citrate to acetate dialysate in patients undergoing HF-HD and HDF found no differences in KT (53 vs. 53.9) [18] or Kt/V (1.49 ± 0.06 vs. 1.47 ± 0.07) [9]. Similarly, a study on HF-HD showed no differences in Kt/V between the dialysates (1.52 ± 0.37 vs. 1.53 ± 0.31) [42]. In contrast, a 2000 study on HF-HD observed a significant increase in Kt/V (1.23 ± 0.19 to 1.34 ± 0.20, *p* = 0.01) [13], as did a study involving HDF patients, favoring citrate (58.44 ± 3.37 vs. 56.94 ± 3.18) [43]. A 2005 report from the same group noted that citrate’s anticoagulant effect allowed for dialyzer reuse [16]. Our findings align with most of the published research, indicating no significant difference in the clearance of small molecules like urea and creatinine, resulting in a similar dialysis dose by standard definitions.

Notably, we found a small difference in the depuration of the uremic toxin p-cresyl sulfate. P-cresyl and indoxyl-sulfate are organic compounds produced in the large intestine by the fermentation of aerobic gut microbiota that, in the setting of chronic kidney disease, increase the conversion from urea to ammonia in the bowel, therefore increasing the production of these toxins [32]. Though small, these uremic toxins are difficult to clear with dialysis as more than 80% of them circulate in plasma bound to proteins [33,34,44]. Albumin has a low-affinity and a high-affinity binding site for these toxins [37], and their clearance could be enhanced by increasing competition for this binding site during dialysis. The clearance of p-cresyl in patients undergoing hemodialysis is crucial due to its harmful effects on various biological systems [45]. Elevated levels have been linked to endothelial damage [46], oxidative stress, and vascular remodeling [47], all of which contribute to cardiovascular complications. It also promotes the production of reactive oxygen species (ROS) in cardiomyocytes, leading to cardiac apoptosis and diastolic dysfunction [48]. Furthermore, p-cresyl is associated with insulin resistance, abnormal fat metabolism [49], and impaired immune function, as it suppresses the activity of immune cells and decreases phagocyte function [50,51]. These toxic effects highlight the importance of reducing p-cresyl levels to prevent cardiovascular morbidity, mortality, and other severe complications in patients with chronic kidney disease. One of the strategies to increase the clearance of these molecules is to displace them from albumin, raise their free fraction, and allow them to diffuse through dialyzer pores. Several medical treatments have been proposed as displacers, such as furosemide, tryptophan, and ibuprofen [35,36]. In the case of this study, the slight decrease in p-cresyl sulfate clearance with the low calcium and standard citrate dialysate could be due to the reduced displacement from albumin caused by less available calcium or magnesium interfering with the p-cresyl sulfate’s albumin binding site. However, the fact that this was not observed with indoxyl sulfate may suggest that indoxyl sulfate’s binding site may not be affected by calcium or magnesium or that our results may have been due to randomness. 

This study has many limitations. It is a short and small study. We did not measure middle-sized molecules such as β2 microglobulin, which could provide further information on the clearance disparities between dialysates. Also, more mechanistic studies are needed to understand the differences observed in our results with p-cresyl sulfate fully.

## 4. Conclusions

There are no differences in dialysis dose per Kt measurements in patients who receive a chronic dialysis treatment with the HDF modality. However, its use may decrease the plasma concentration of ionized divalent cations (i.e., calcium and magnesium) and reduce the albumin displacement of p-cresyl sulfate. Therefore, until further studies are conducted, we suggest that when citrate dialysate is used, it should be used with a 0.75 mmol/L magnesium concentration rather than a 0.5 mmol/L concentration.

## 5. Materials and Methods

This is a single-center prospective crossover study. It included prevalent patients (defined by a dialysis vintage of at least three months) who participated in a thrice-weekly online HDF program, had less than 250 mL of urine output per day, had a vascular access capable of >350 mL/min of blood flow, and provided informed consent. We excluded patients who were expecting to receive a living donor kidney transplant within the next month, those who had severe (<7.0 mg/dL) or symptomatic hypocalcemia, or had an active infection or neoplastic process. This study was approved by the local Ethics Committee and was conducted according to the Declaration of Helsinki.

Four dialysates were used in this study. Two were acetate-based: SmartBag^®^ 211.25 and SmartBag^®^ 211.5 with a calcium concentration of 1.25 and 1.5 mmol/L, respectively. The remaining two were citrate-based: SmartBag^®^ CA 211.5 and SmartBag^®^ CA 211.5-0.75 with a magnesium concentration of 0.5 and 0.75 mmol/L, respectively, but with 1.5 mmol/L calcium in both cases. Fresenius Medical Care, Bad Homburg, Germany, manufactured the dialysates used. A previously published article describes details of each dialysate’s composition and patient assignment (Table 4) [52]. The rest of the dialysis prescription parameters were not altered during the study, ensuring that any observed effects could be attributed solely to the differences in the dialysates used.

Pre-filter blood samples (10 mL) were taken at the beginning and end of the second weekly treatment session. The dialysis parameters recorded for each session included actual duration, type of dialyzer, blood flow rate (Qb), recirculation index measured by the temperature module, arterial and venous pressures, transmembrane pressure (TMP), initial and final hematocrit automatically measured by the Blood Volume Monitor (BVM)^®^ biosensor, Fresenius Medical Care, Bad Homburg, Germany, initial and final body weights, the volume of blood processed, and replacement volume.

Laboratory measurements included determining the concentrations of various solutes in the serum at the start and end of each session to determine their reduction ratio expressed as a percentage. These solutes were urea (60 Da), creatinine (113 Da), p-cresyl (108 Da), indoxyl sulfate (213 Da), and albumin (67 kDa). Indoxyl sulfate and p-cresyl sulfate were measured in serum using liquid chromatography–mass spectrometry (LC-MS). Briefly, isotopically labeled internal standards were added to 50 µL of serum and 1 mL of methanol. After protein precipitation and evaporation of the organic phase, samples were re-dissolved in 100 µL of water and injected into an LC-MS Orbitrap Exploris 120 instrument, Thermo Fisher Scientific Inc ©, Waltham, MA USA. The mobile phase consisted of water (+0.1% formic acid) and methanol (+0.1% formic acid). The accuracy and precision of the method were <15%.

The final concentrations of p-cresyl sulfate, indoxyl sulfate, and albumin were adjusted for hemoconcentration and volume of distribution (approximate extracellular volume) according to the formula described by Bergström and Wehle [53]. 

The one-sample Kolmogorov–Smirnov test was used to test the distribution of the studied variables. The results are expressed as the arithmetic mean ± standard deviation. For the analysis of the statistical significance of quantitative parameters, a paired Student *t*-test was used to compare citrate vs. acetate. Repeated measures of ANOVA or Friedman’s test for non-parametric variables were employed to analyze data from the four different dialysates. Bonferroni’s post hoc tests were performed for pairwise comparisons; *p*-values < 0.05 were considered statistically significant. Analyses were performed using SPSS software, version 23 (SPSS Inc., Chicago, IL, USA).

## Figures and Tables

**Figure 1 toxins-16-00426-f001:**
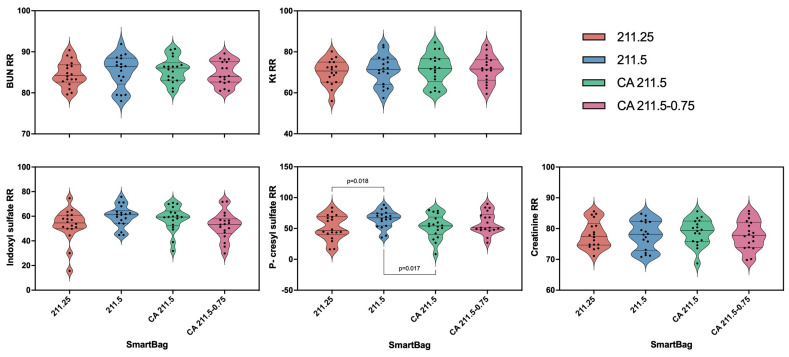
Comparison of reduction ratios (RRs) for the studied uremic toxins between dialysates. ANOVA for repeated measures showed statistically significant differences between groups in p-cresyl sulfate RR; Bonferroni corrections results are represented. No statistically significant differences were found between the other RRs.

**Table 1 toxins-16-00426-t001:** Dialysis dose and reduction ratios for dialyzable and protein-bound small molecules with the use of acetate vs. citrate as dialysates.

Variable	Acetate	Citrate	*p*-Value
Kt (L, mean ± SD)	70.52 ± 6.58	71.56 ± 6.88	0.087
BUN RR (%, mean ± SD)	84.93 ± 3.53	85.19 ± 2.98	0.489
Creatinine RR (%, mean ± SD)	77.95 ± 4.5	78.4 ± 4.44	0.188
P-cresyl sulfate RR (%, mean ± SD)	58.4 ± 18.58	55.84 ± 18.39	0.449
Indoxyl-sulfate RR (%, mean ± SD)	56.39 ± 11.6	55.11 ± 10.96	0.56

BUN, blood urea nitrogen; RR, reduction ratio; and SD, standard deviation.

**Table 2 toxins-16-00426-t002:** Dialysis dose and reduction ratios for dialyzable and protein-bound small molecules with acetate and citrate dialysates with different calcium and magnesium concentrations.

Variable	SmartBag 211.25	SmartBag 211.5	SmartBag CA 211.5	SmartBag CA 211.5-0.75	*p*-Value
Kt (L, mean ± SD)	70.7 ± 1.44	70.97 ± 1.70	71.64 ± 1.77	71.48 ± 1.52	0.353
BUN RR (%, mean ± SD)	84.62 ± 0.76	85.23 ± 0.94	85.60 ± 0.72	84.79 ± 0.69	0.347
Creatinine RR (%, mean ± SD)	78.05 ± 1.05	77.84 ± 1.1	78.89 ± 0.99	77.9 ± 1.11	0.273
p-cresyl sulfate RR (%, mean ± SD)	51.56 ± 4.75	65.25 ± 3.38	53.02 ± 4.52	58.66 ± 4.16	0.007
Indoxyl-sulfate RR (%, mean ± SD)	52.84 ± 3.1	58.94 ± 2.07	57.87 ± 2.43	52.35 ± 2.64	0.063

BUN, blood urea nitrogen; RR, reduction ratio; SmartBag 211.25, calcium 1.25 mmol/L and magnesium 0.5 mmol/L; SmartBag 211.5, calcium 1.5 mmol/L and magnesium 0.5 mmol/L; SmartBag CA 211.5, calcium 1.5 mmol/L and magnesium 0.5 mmol/L; SmartBag CA 211.5-0.75, calcium 1.5 mmol/L and magnesium 0.75 mmol/L; and SD, standard deviation.

**Table 3 toxins-16-00426-t003:** Post hoc analysis of the p-cresyl reduction ratio by analyzing all four dialysates against each other.

Dialysate	Adjusted Marginal Mean % (95% CI)	Mean Difference ± SD	Bonferroni-Adjusted CI	*p*-Value
SmartBag 211.5	65.25 (58.12, 72.38)			
SmartBag 211.25		13.69 ± 3.97	1.86, 25.53	0.018
SmartBag CA 211.5		12.23 ± 3.51	1.76, 22.7	0.017
SmartBag CA 211.5-0.75		6.59 ± 3.47	−3.75, 16.94	0.45
SmartBag 211.25	51.56 (41.53, 61.58)			
SmartBag 211.5		−13.69 ± 3.97	1.86, 25.53	0.018
SmartBag CA 211.5		−1.47 ± 4.23	−14.08, 11.14	1
SmartBag CA 211.5-0.75		−7.1 ± 4.78	−21.36, 7.16	0.93
SmartBag CA 211.5	53.02 (43.49, 62.56)			
SmartBag 211.5		−12.23 ± 3.51	−22.7, −1.76	0.017
SmartBag 211.25		1.47 ± 4.23	−11.14, 14.08	1
SmartBag CA 211.5-0.75		−5.63 ± 4.85	−20.1, 8.83	1
SmartBag CA 211.5-0.75	58.65 (49.87, 67.44)			
SmartBag 211.5		−6.59 ± 3.47	−16.94, 3.75	0.45
SmartBag 211.25		7.1 ± 4.78	−7.16, 21.36	0.93
SmartBag CA 211.5		5.63 ± 4.85	−8.83, 20.1	1

CI, confidence interval; SD, standard deviation; SmartBag 211.25, calcium 1.25 mmol/L and magnesium 0.5 mmol/L; SmartBag 211.5, calcium 1.5 mmol/L and magnesium 0.5 mmol/L; SmartBag CA 211.5, calcium 1.5 mmol/L and magnesium 0.5 mmol/L; SmartBag CA 211.5-0.75, calcium 1.5 mmol/L and magnesium 0.75 mmol/L; and SD, standard deviation.

**Table 4 toxins-16-00426-t004:** Dialysates’ composition.

Components	SmartBag 211.25	SmartBag 211.5	SmartBagCA 211.5	SmartBagCA 211.5-0.75
Sodium (mmol/L)	138	138	138	138
Potassium (mmol/L)	2	2	2	2
Calcium (mmol/mL)	1.25	1.5	1.5	1.5
Magnesium (mmol/mL)	0.5	0.5	0.5	0.75
Chloride (mmol/mL)	108.5	109	109	109.5
Acetate (mmol/L)	3	3	-	-
Citrate (mmol/L)	-	-	1	1
Glucose (g/L)	1	1	1	1
Bicarbonate (mmol/L)	32	32	32	32
Osmolarity (mosm/L)	290.8	291.55	290	290

## Data Availability

The data supporting the findings of this study are available on GitHub (https://github.com/Broseta/Citrate-dialysate.git, accessed on 15 August 2024).

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
