# Peer review of "Comparative Effects of Acetate- and Citrate-Based Dialysates on Dialysis Dose and Protein-Bound Uremic Toxins in Hemodiafiltration Patients: Exploring the Impact of Calcium and Magnesium Concentrations"

_toxins, 2024, doi:10.3390/toxins16100426_

Round 1
Reviewer 1 Report
Comments and Suggestions for Authors
The manuscript describes perliminary regarding the influence of the type of dialysate used in HD.
the article is weel structured, yet the discussion and conclusions could be reformulated and enfatize the importance of this increment in the RR of p-cresyl sulfate.
Also, the formulas and abbreviations should be revised.
Author Response
Comment 1: The manuscript describes perliminary regarding the influence of the type of dialysate used in HD.
The article is weel structured, yet the discussion and conclusions could be reformulated and enfatize the importance of this increment in the RR of p-cresyl sulfate.
Response 1: Thank you for your comments. We have taken your suggestions and added the following to the p-cresyl paragraph in the discussion section:
“The clearance of p-cresyl in patients undergoing hemodialysis is crucial due to its harmful effects on various biological systems [47]. Elevated levels have been linked to endothelial damage [48], oxidative stress, and vascular remodeling [49], all of which contribute to cardiovascular complications. It also promotes the production of reactive oxygen species (ROS) in cardiomyocytes, leading to cardiac apoptosis and diastolic dysfunction [50]. Furthermore, p-cresyl is associated with insulin resistance, abnormal fat metabolism [51], and impaired immune function, as it suppresses the activity of immune cells and decreases phagocyte function [52, 53]. These toxic effects highlight the importance of reducing p-cresyl levels to prevent cardiovascular morbidity, mortality, and other severe complications in patients with chronic kidney disease.”
Comment 2: Also, the formulas and abbreviations should be revised.
Response 2: Thank you for pointing out these mistakes. We have corrected the chemical formulas to display the number of atoms as subscripts. Additionally, we have reviewed and corrected abbreviations that were previously undefined, such as AVF, BVM, HDF, and HF-HD.
Reviewer 2 Report
Comments and Suggestions for Authors
In the present study the main focus of the authors was to address the possible benefits of the administration of divalent cations ( calcium [Ca] or magnesium [Mg]) during hemodiafiltration using acetate or citrate as buffer on the removal of uremic toxins, specifically para-cresyl sulfate and indoxyl sulfate. The authors data although limited to studies of only 18 patients seem to indicate first that the only uremic toxin favorably influenced ( greater removal) by this manoeuvre was para cresyl sulfate ( but not indoxyl sulfate) and secondly that the fluids containing a higher concentration of Calcium and an higher concentration of magnesium had the greatest benefit to enhance toxin removal.
The authors state that they did a cross over study; did they mean that the patients crossed over from one acetate bag to another with a different calcium concentration and from one citrate bag to another with a different Magnesium concentration or was it a four-way cross over. In whichever manner they conducted the study then the comparison should be:
1) Acetate with higher calcium vs. lower calcium
2) Citrate with higher magnesium vs. lower
3) Before looking at differences between the two different buffer groups – acetate vs. citrate.
This was not apparent from the RESULTS section
As the premise is that acetate and citrate, by virtue of their affinity for calcium and magnesium, clear binding site on albumin previously occupied by these cations, allowing the uremic toxins to bind to albumin, the inclusion of calcium and magnesium in the fluid will to the albumin resulting in more free uremic toxins available for diffusion. Not surprisingly the authors found that the incorporation of calcium and magnesium, especially in higher concentrations, enhanced the removal of paracresyl sulfate (though not indoxyl sulfate) . This observation is of significant clinical value – as the authors indicate and recommend its implementation.
Comments on the Quality of English Languageline 99 the indoxyl sulfate pre dialysis values are not completed nor is it a sentence.
Importantly the first sentence of the discussion line122-124 contains a double negative which makes it virtually incomprehensible.
English in the subsequenty paragraph of the discussion lines 126-134 should be improved. it is a litany of "and another...."
Author Response
Comment 1: In the present study the main focus of the authors was to address the possible benefits of the administration of divalent cations (calcium [Ca] or magnesium [Mg]) during hemodiafiltration using acetate or citrate as buffer on the removal of uremic toxins, specifically para-cresyl sulfate and indoxyl sulfate. The authors data although limited to studies of only 18 patients seem to indicate first that the only uremic toxin favorably influenced (greater removal) by this manoeuvre was para cresyl sulfate (but not indoxyl sulfate) and secondly that the fluids containing a higher concentration of Calcium and an higher concentration of magnesium had the greatest benefit to enhance toxin removal.
The authors state that they did a cross over study; did they mean that the patients crossed over from one acetate bag to another with a different calcium concentration and from one citrate bag to another with a different Magnesium concentration or was it a four-way cross over. In whichever manner they conducted the study then the comparison should be:
1) Acetate with higher calcium vs. lower calcium
2) Citrate with higher magnesium vs. lower
3) Before looking at differences between the two different buffer groups – acetate vs. citrate.
This was not apparent from the RESULTS section.
As the premise is that acetate and citrate, by virtue of their affinity for calcium and magnesium, clear binding site on albumin previously occupied by these cations, allowing the uremic toxins to bind to albumin, the inclusion of calcium and magnesium in the fluid will to the albumin resulting in more free uremic toxins available for diffusion. Not surprisingly the authors found that the incorporation of calcium and magnesium, especially in higher concentrations, enhanced the removal of paracresyl sulfate (though not indoxyl sulfate) . This observation is of significant clinical value – as the authors indicate and recommend its implementation.
Response 1: Thank you for your suggestions. We have performed further analysis to confront every data from every dialysate against each other and have added the results both in table and in paragraph form in the text:
“Given the significant difference in the p-cresyl results, we further analyzed this variable by examining the adjusted marginal means differences and found significant ones between SmartBag® 211.5 with SmartBag® 211.25 and SmartBag® CA 211.5. While there were no significant differences when compared to SmartBag® CA 211.5-0.75 (see Table 3 for further information).”
Table 3. Post-hoc analysis of the p-cresyl reduction ratio by analyzing all four dialysates against each other.
|
Dialysate |
Adjusted marginal mean % (95% CI) |
Mean difference ± SD |
Bonferroni- adjusted CI |
p-value |
|
SmartBag 211.5 |
65.25 (58.12, 72.38) |
|
|
|
|
SmartBag 211.25 |
|
13.69 ± 3.97 |
1.86, 25.53 |
0.018 |
|
SmartBag CA 211.5 |
|
12.23 ± 3.51 |
1.76, 22.7 |
0.017 |
|
SmartBag CA 211.5-0.75 |
|
6.59 ± 3.47 |
-3.75, 16.94 |
0.45 |
|
SmartBag 211.25 |
51.56 (41.53, 61.58) |
|
|
|
|
SmartBag 211.5 |
|
-13.69 ± 3.97 |
1.86, 25.53 |
0.018 |
|
SmartBag CA 211.5 |
|
-1.47 ± 4.23 |
-14.08, 11.14 |
1 |
|
SmartBag CA 211.5-0.75 |
|
-7.1 ± 4.78 |
-21.36, 7.16 |
0.93 |
|
SmartBag CA 211.5 |
53.02 (43.49, 62.56) |
|
|
|
|
SmartBag 211.5 |
|
-12.23 ± 3.51 |
-22.7, -1.76 |
0.017 |
|
SmartBag 211.25 |
|
1.47 ± 4.23 |
-11.14, 14.08 |
1 |
|
SmartBag CA 211.5-0.75 |
|
-5.63 ± 4.85 |
-20.1, 8.83 |
1 |
|
SmartBag CA 211.5-0.75 |
58.65 (49.87, 67.44) |
|
|
|
|
SmartBag 211.5 |
|
-6.59 ± 3.47 |
-16.94, 3.75 |
0.45 |
|
SmartBag 211.25 |
|
7.1 ± 4.78 |
-7.16, 21.36 |
0.93 |
|
SmartBag CA 211.5 |
5.63 ± 4.85 |
-8.83, 20.1 |
1 |
CI, confidence interval; SD, standard deviation; SmartBag 211.25, calcium 1.25 mmol/L and magnesium 0.5 mmol/L; SmartBag 211.5, calcium 1.5 mmol/L and magnesium 0.5 mmol/L; SmartBag CA 211.5, calcium 1.5 mmol/L and magnesium 0.5 mmol/L; SmartBag CA 211.5-0.75, calcium 1.5 mmol/L and magnesium 0.75 mmol/L; SD, standard deviation.
Comment 2: line 99 the indoxyl sulfate pre dialysis values are not completed nor is it a sentence.
Response 2: You are correct, thank you for noticing this. This typo has been fixed.
“There were no statistically significant differences in the p-cresyl sulfate median pre-dialysis values between SmartBag® 211.25 (43308 ng/mL, 32095 – 80991 ng/mL), 211.5 (56728 ng/mL, 42578 – 89815 ng/mL), CA 211.5 (50074 ng/mL, 39417 – 76838 ng/mL), and CA 211.5-0.75 (54813 ng/mL, 35332 – 80281 ng/mL) (p= 0.372); nor in the indox-yl-sulfate median pre-dialysis values between SmartBag® 211.25 (51024 ng/mL, 38264 – 63785 ng/mL), 211.5 (53004 ng/mL, 41769 - 64240 ng/mL), CA 211.5 (55615 ng/mL, 42583 - 68648 ng/mL) and CA 211.5-0.75 (53587 ng/mL, 42435 - 64740 ng/mL) (p= 0.6).”
Comment 3: Importantly the first sentence of the discussion line122-124 contains a double negative which makes it virtually incomprehensible.
Response 3: We understand the confusion and given the new performed analysis we have changed the first statement to the following:
“There was a significantly greater reduction ratio of p-cresyl sulfate when patients used the acetate dialysate with a standard calcium concentration of 1.5 mmol/L than with 1.25 mmol/L or with the citrate dialysate with a calcium concentration of 1.5 mmol/L and a magnesium concentration of 0.5 mmol/L. There were no significant differences when compared to the magnesium supplemented citrate dialysate (0.75 mmol/L).”
Comment 4: English in the subsequenty paragraph of the discussion lines 126-134 should be improved. it is a litany of "and another...."
Response 4: We appreciate your insight. We have improved the English in this paragraph to the following:
“Several studies have analyzed the depurative capacity of citrate vs. acetate, yielding mixed results. Two studies comparing citrate to acetate dialysate in patients undergoing HF-HD and HDF found no differences in KT (53 vs. 53.9) [18] or Kt/V (1.49 ± 0.06 vs. 1.47 ± 0.07) [9]. Similarly, a study on HF-HD showed no differences in Kt/V between the dialysates (1.52 ± 0.37 vs. 1.53 ± 0.31) [44]. In contrast, a 2000 study on HF-HD observed a significant increase in Kt/V (1.23 ± 0.19 to 1.34 ± 0.20, p=0.01) [44], as did a study involving HDF patients, favoring citrate (58.44 ± 3.37 vs. 56.94 ± 3.18) [45]. A 2005 report from the same group noted that citrate’s anticoagulant effect allowed for dialyzer reuse [16]. Our findings align with most of the published research, indicating no significant difference in the clearance of small molecules like urea and creatinine, resulting in a similar dialysis dose by standard definitions.”